# Nanoindentation of γ-TiAl with Different Crystal Surfaces by Molecular Dynamics Simulations

**DOI:** 10.3390/ma12050770

**Published:** 2019-03-06

**Authors:** Xiaocui Fan, Zhiyuan Rui, Hui Cao, Rong Fu, Ruicheng Feng, Changfeng Yan

**Affiliations:** 1School of Mechanical and Electronical Engineering, Lanzhou University of Technology, Lanzhou 730050, China; xiaoc_fan@163.com (X.F.); c_hui0722@126.com (H.C.); f_rong0901@126.com (R.F.); frcly@163.com (R.F.); changf_yan@163.com (C.Y.); 2State Key Laboratory of Advanced Processing and Recycling of Non-ferrous Metals, Lanzhou University of Technology, Lanzhou 730050, China

**Keywords:** nanoindentation, γ-TiAl, crystal orientation, molecular dynamics simulation

## Abstract

The periodicity and density of atomic arrangement vary with the crystal orientation, which results in different deformation mechanisms and mechanical properties of γ-TiAl. In this paper, the anisotropic characteristics for γ-TiAl with (100), ($\bar{1}10$) and (111) surfaces during nanoindentation at 300 K have been investigated by molecular dynamics simulations. It is found that there is no obvious pop-in event in all load-depth curves when the initial plastic deformation of γ-TiAl samples occurs, because the dislocation nucleates before the first load-drop; while a peak appears in both the unloading curves of the ($\bar{1}10$) and (111) samples due to the release of energy. Stacking faults, twin boundaries and vacancies are formed in all samples; however, interstitials are formed in the (100) sample, a stacking fault tetrahedron is formed in the (111) sample; and two prismatic dislocation loops with different activities are formed in the ($\bar{1}10$) and (111) samples, respectively. It is also concluded that the values of the critical load, strain energy, hardness and elastic modulus for the (111) sample are the maximum, and for the (100) sample are the minimum. Furthermore, the orientation dependence of the elastic modulus is greater than the hardness and critical load.

## 1. Introduction

Currently, micro–electromechanical systems (MEMS) have been successfully applied in fields such as biology or aerospace, which means that components of MEMS should be suitable for an increasingly complex working environment. As an intermetallic compound material, γ-TiAl has the high temperature strength and specific modulus, excellent oxidation resistance and flame retardance, low density and expansion coefficient, these advantages give it great application potential in the field of MEMS [1,2,3,4].

Since the MEMS devices can be manufactured at the nanoscale, the mechanical properties of nanomaterials show great significance for their application. Nanoindentation is an effective method to measure the mechanical properties of materials at the nanoscale, such as the elastic modulus, hardness and strain hardening effect. So far, this technology has been applied to obtain the mechanical properties of various materials including not only pure metals such as Al, Cu, Ag, Ni, and Fe [5,6,7,8]; but also alloys such as TiAl, U–Cu, Fe–Ni–C, Zr–Cu–Ag–Al and nickel superalloys [9,10,11,12,13,14,15].

However, it is difficult to investigate the transient atomic information inside the materials during nanoindentation experiments [16,17]. As a powerful supplement to the experiments, the molecular dynamics method can simulate the atomic–scale interaction between the indenter and materials, analyze the defect evolution, and then obtain the mechanical properties and deformation mechanism in detail [18]. Liu and Jiao et al. [19,20] studied the formation mechanism of prismatic dislocation loop and stacking fault tetrahedron during the nanoindentation of Al and Cu, respectively. Shih–Wei and Talaei et al. [21,22] investigated the grain boundary effects on the indentation-induced plastic deformation of Cu and Fe. Abu-Shams and Biao et al. [23,24] revealed the effect of void on the nanoindentation of Fe–10% Cr and Ni alloy. Furthermore, Dasilva et al. [25] found that both emission and interaction of dislocations in γ-TiAl were mediated by the expansion of glide loops on the {111} planes resulting in the formation of prismatic loops. Xu et al. [26] observed that the critical load increased with the indentation depth, while the indentation speeds of 0.001–0.15 Å/ps had little influence on Young’s modulus of γ-TiAl.

In general, the periodicity and density of atomic arrangement vary with the crystal orientation, which results in different physical properties of crystals. Jun et al. [27] performed nanoindentation tests on dual-phase Ti alloys and found that the grain orientation had a significant influence on the local strain rate sensitivity of Ti6242. Ziegenhain et al. [28] carried out the nanoindentation simulations of Cu and Al, the result showed that the influence of the surface orientation was lost when the plastic deformation had set in. Xiong et al. [29,30] found that the critical load, dislocation nucleation position and critical contact pressure were associated with the crystal orientation of Ni_3_Al and FeNi_3_. Kempf et al. [31] indicated that the different gamma domains had little influence on the hardness of twinned TiAl with (110) and (111) gamma by nanoindentation experiment, while the plastic anisotropy caused significant differences in the pile-up. Zambaldi et al. [32] studied the plastic anisotropy of γ-TiAl and confirmed that the easy activation of ordinary dislocation glided in stoichiometric γ-TiAl. In addition, the lattice instability of γ-TiAl at 0.1 K was investigated by nanoindentation simulation, it was found that the orientation of crystal affected the stress field and the nucleation of dislocation [16].

Although there are several works about γ-TiAl with different surfaces, the effect of crystal orientation on the deformation mechanisms and mechanical properties of γ-TiAl at room temperature is still not sufficiently clear. In order to supply the current inadequacy, in this paper, LAMMPS [33] is used to simulate the nanoindentation processes (loading and unloading) of γ-TiAl with (100), ($\bar{1}10$) and (111) surfaces at 300 K. The defect evolution, load-depth curve, hardness, elastic modulus, critical load and strain energy for different samples are analyzed in detail.

## 2. Materials and Methods

For each case in the present study, as illustrated in Figure 1, the nanoindentation model consists of the indenter and substrate. The indenter is a diamond sphere with a radius of 28 Å, which is regarded as a rigid body, and its tip is positioned 7.4 Å apart from the crystal surface at first. The size of γ-TiAl substrate is 220 Å × 220 Å × 220 Å, including the boundary layer, thermostat layer and Newtonian layer. The thickness of the boundary layer is 10 Å, atoms in this region are fixed to prevent the substrate from moving. The thermostat layer using the velocity rescaling method helps to control the temperature of the whole system, its thickness is 30 Å. The thickness of the Newtonian layer is 180 Å, atoms in this region obey Newton’s law of motion. The interaction of Al-Ti is described by the embedded atom method (EAM) potential [34]. The interaction of C-C is neglected because the indenter is rigid. The interactions of C-Al and C-Ti are described by the Mie 6–12 potential (often called the Lennard–Jones potential) [35]:(1)$$U(r)=4\epsilon \left[{\left(\frac{\sigma }{r}\right)}^{12}-{\left(\frac{\sigma }{r}\right)}^{6}\right],r\;<\;r_{0}$$
where *ϵ* is the depth of potential well, *σ* is the equilibrium distance, and *r*_0_ is the cutoff distance. These parameters are listed in Table 1 [27,36], and the cutoff distance is 2.5 times the equilibrium distance [37].

Owing to the limitation of computing power, the loading speed in molecular dynamics simulations is generally in the range of 1–100 m/s [38], although which is much higher than the 10^−6^–10^−9^ m/s in nanoindentation experiments [39,40]. In this paper, a speed of 50 m/s is applied along the direction of –z, and that was also chosen in other studies [41,42]. The time step used is 1 fs, and the maximal indentation depth is 22.6 Å. The free boundary condition is applied along the z-direction, and periodic boundary conditions are applied along the x and y directions to eliminate the effect of boundary conditions on the side faces. Besides, the initial relaxation was performed properly for 380 ps at 300 K before indentation to make the system to be in equilibrium.

The crystal structures of different γ-TiAl samples are displayed in Figure 2, the base vectors x, y and z are set as [$00\bar{1}$], [010] and [100] for the (100) sample (Figure 2a); [$\bar{1}\bar{1}2$], [111] and [$\bar{1}10$] for the ($\bar{1}10$) sample (Figure 2b); and [$\bar{1}\bar{1}2$], [$1\bar{1}0$] and [111] for the (111) sample (Figure 2c).

## 3. Results

### 3.1. Analysis of Load-Depth Curves

The load-depth curves with characteristic marks are depicted in Figure 3. The repulsive force is positive and the attractive force is negative. The summation of the vertical components of the force exerted on the indenter by the substrate is considered as the load. The force can be obtained by the derivative of the Mie 6–12 (Lennard–Jones) potential function.

For each case, the whole nanoindentation procedure can be divided into four stages as follows: stage I, the indenter is approaching the substrate. When the indenter is positioned about 2 Å apart from the substrate, there is an attraction between the indenter and the substrate surface. However, the attraction is too small to make the atoms of the substrate surface leave their original positions and adhere to the indenter. Stage II, the indenter is pressing into the substrate. A repulsive force exists between the indenter and substrate and increases with the indentation depth. Stage III, the indenter is returning and the load is positive. The indenter begins to return after reaching the maximum depth, and the load decreases as the indenter returns. Due to the plastic deformation occurring in the γ-TiAl substrate when the load decreases to zero, the indenter cannot return to its original position. Stage IV, the indenter continues to move back while the load is not positive. In the meantime, there is only an attraction between the indenter and γ-TiAl substrate before the load is zero, and then the indenter returns to its original position gradually.

During each loading process, several pop-in events are observed in the load-depth curve, which can be attributed to the release of strain energy accumulated during the deformation through dislocation activities. It is generally believed that the first pop-in event represents the occurrence of initial plastic deformation [43]. However, because the dislocation nucleates prior to the first load drop, there is no pronounced pop-in event in the load-depth curve when the initial plastic deformation occurs in this study; in the nanoindentations of Ni_3_Al and Co, similar behaviors were observed [29,44].

During the unloading process, peaks H and G appear in the load-depth curves of the ($\bar{1}10$) and (111) samples, respectively. This is associated with the dislocation annihilation that will cause the release of energy.

The critical load of the transformation of elastic–plastic deformation is the force exerted by the indenter on the sample when the initial dislocation nucleates. The critical depth of the initial dislocation nucleation can be obtained by the Ovito, and the critical load can be obtained by the critical depth from the load-depth curves in Figure 3. Note that the critical load is 36.71 ev/Å of the (100) sample, 41.38 ev/Å of the ($\bar{1}10$) sample and 53.29 ev/Å of the (111) sample. Obviously, the value of the (100) sample is the minimum, and that of the (111) sample is the maximum. The results are consistent with the previous study [16]. It indicates that the resistance to plastic deformation of (111) sample is the strongest, while that of the (100) sample is the weakest.

### 3.2. Analysis of Defect Evolution

In this work, the centrosymmetry parameter (CSP) [45] and dislocation extraction algorithm (DXA) [36] are used to visualize and identify interior defects in samples during nanoindentation. Figure 4, Figure 5 and Figure 6 present a series of snapshots exhibiting the defect evolution in different γ-TiAl samples, corresponding to the characteristic points marked in the load-depth curves of Figure 3. The Other atoms are coloured white, the HCP atoms are coloured pink and the FCC atoms are coloured green. The Stair-rod dislocation lines are coloured purple, the Hirth dislocation lines are coloured yellow, the Perfect dislocation lines are coloured blue and the Shockley dislocation lines are coloured green. Besides, a single HCP atom layer represents a twin boundary, two adjacent HCP atom layers present an intrinsic stacking fault, and an FCC atom layer in the middle of two HCP atom layers stands for an intrinsic stacking fault.

Initially, γ-TiAl deforms elastically without defect nucleation. With the increasing indentation depth, the substrate atoms beneath the indenter deviate from their original positions and arrange disorderly, the strain energy increases, and then the dislocation nucleates causing plastic deformation. Because the dislocation sources exist in γ-TiAl, the number of dislocations increases with the indentation depth. The dislocation multiplication is dominated by the Shockley dislocation, which can improve the slip ability of the sample that intensifies the plastic deformation.

#### 3.2.1. Analysis of Defect Evolution for the (100) Sample

In Figure 4a, when the indentation depth is 3.2 Å (corresponding to A in Figure 3a), a dislocation embryo nucleates below the top surface, then the sample enters the plastic deformation stage. Subsequently, the atoms around the dislocation embryo gradually deviate from their equilibrium positions and develop into a dislocation. Dislocations glide with the increasing indentation depth, which may destroy the normal periodic arrangement of atoms and results in the formation of stacking faults (SFs). As shown in Figure 4b, as the depth increases to 7.45 Å (corresponding to B in Figure 3a), an intrinsic stacking fault (SISF) on the $(1\bar{1}1$) plane is formed; moreover, a 1/2[$0\bar{1}\bar{1}$] superpartial dislocation dissociates into 1/6[$1\bar{2}\bar{1}$] and 1/6[$\bar{1}\bar{1}\bar{2}$] Shockley dislocations connected by an SISF on the ($11\bar{1}$) plane; a 1/2[$0\bar{1}1$] superpartial dislocation dissociates into 1/6[$\bar{1}\bar{1}2$] and 1/6[$1\bar{2}1$] Shockley dislocations connected by an SISF on the (111) plane. The dislocation dissociations occur due to the low enough SF energies [46].

With the further increasing indentation depth, the stacking faults grow because of various dislocation activities. In Figure 4c, as the depth reaches 15.9 Å (corresponding to C in Figure 3a), the previous ($11\bar{1}$) and (111) SISFs disappear with dislocation reactions, and new SISFs are formed on the ($11\bar{1}$) and ($1\bar{1}1$) planes. An extended dislocation is formed due to the dissociation of 1/2[$1\bar{1}0$] Perfect dislocation into 1/6[$1\bar{2}\bar{1}$] and 1/6[$2\bar{1}1$] Shockley dislocations connected by an SISF on the ($11\bar{1}$) plane. Furthermore, an extrinsic stacking fault (SESF) appears on the ($11\bar{1}$) plane (Figure 4c_0_), which is caused by a layer of atoms inserting in the normal stacking sequence. As Figure 4d shows, when the depth increases to 20.35 Å (corresponding to D in Figure 3a), the ($11\bar{1}$) SESF and the extended dislocation fade away; while fresh SISFs are formed on the (111), ($1\bar{1}1$) and ($\bar{1}11$) planes; what’s more, a Stair-rod dislocation, which is immovable and stable, is generated by the dislocation reaction 1/6[$\bar{2}\bar{1}\bar{1}$] + 1/6[121] → 1/6[$\bar{1}10$].

As displayed in Figure 4e, when the depth reaches the maximum depth of 22.6 Å (corresponding to E in Figure 3a), there are still the (111) and ($1\bar{1}1$) SISFs; while a twinning boundary (TB) on the ($11\bar{1}$) plane is formed (Figure 4e_0_), and a similar behavior was observed during the nanoindentation of Ag with the (001) surface [47]. In addition, the previous Stair-rod dislocation annihilates, and a fresh 1/6[$0\bar{1}1$] Stair-rod dislocation is formed by the dislocation reaction 1/6[$1\bar{2}\bar{1}$] + 1/6[$\bar{1}12$] → 1/6[$0\bar{1}1$]. Afterwards, with the constant rise of the indenter, dislocations will move toward the indentation and shrink, even become annihilated. In Figure 4f, as the indentation depth decreases to 14.1 Å (corresponding to F in Figure 3a), there is no dislocation in the sample, while ten vacancies and four interstitials are formed due to dislocation reactions. As Figure 4g shows, the indenter returns to the original position (corresponding to G in Figure 3a), vacancies and interstitials are still in the sample, but the number of interstitials decreases because of deformation recovery.

#### 3.2.2. Analysis of Defect Evolution for the ($\bar{1}10$) Sample

As is depicted in Figure 5a, when the indentation depth reaches 3.4 Å (corresponding to A in Figure 3b), a dislocation embryo nucleates and will develop into a 1/6[$11\bar{2}$] Shockley dislocation; meanwhile, the sample deforms plastically. In Figure 5b, as the depth increases to 6.55 Å (corresponding to B in Figure 3b), 1/6[$\bar{1}\bar{1}2$] and 1/6[$\bar{2}11$] Shockley dislocations are formed under indentation. As Figure 5c shows, when the depth reaches 10.95 Å (corresponding to C in Figure 3b), dislocations increase and glide, an SISF is formed on the ($11\bar{1}$) plane, and an SESF is formed on the ($1\bar{1}1$) plane (Figure 5c_0_). In Figure 5d, when the depth is 14.5 Å (corresponding to D in Figure 3b), the ($1\bar{1}1$) SESF annihilates and a TB appears on the ($1\bar{1}1$) plane, which may reduce the energies of planar defects (Figure 5d_0_); meanwhile, there are two SISFs on the ($\bar{1}11$) and ($1\bar{1}1$) planes. Furthermore, a 1/6[$\bar{1}01$] Stair-rod dislocation is formed by the dislocation reaction 1/6[$1\bar{1}2$] + 1/6[$\bar{2}1\bar{1}$] → 1/6[$\bar{1}01$].

In Figure 5e, as the depth is 17.15 Å (corresponding to E in Figure 3b), stacking faults shrink, an ESF (Figure 5e_0_) on the ($11\bar{1}$) plane is formed and the previous ($1\bar{1}1$) TB fades away because of the complex dislocation reactions. As displayed in Figure 5f, when the indentation depth reaches 20.85 Å (corresponding to F in Figure 3b), dislocations interact with each other to form a prismatic dislocation loop, which also appeared in the nanoindentation of the Ni (011) surface [42]. An immovable 1/3[$0\bar{1}0$] Hirth dislocation is formed by the dislocation reaction 1/6[$\bar{1}\bar{1}\bar{2}$] + 1/6[$1\bar{1}2$] → 1/3[$0\bar{1}0$]; the ($11\bar{1}$) SESF fades away accompanying by the formation of an SISF on the (111) plane, and a fresh twin boundary is produced on the ($1\bar{1}1$) plane (Figure 5f_0_).

As shown in Figure 5g, when the indentation depth increases to the maximum 22.6 Å (corresponding to G in Figure 3b), the previous (111) SISF annihilates and a fresh SISF appears on the (111) plane. Moreover, the prismatic dislocation loop develops and its components react subsequently, and a 1/6[$\bar{1}\bar{1}0$] Stair-rod dislocation is formed by the dislocation reaction 1/6[$\bar{2}11$] + 1/6[$1\bar{2}\bar{1}$] → 1/6[$\bar{1}\bar{1}0$]; the growth of the prismatic dislocation loop can result in the release of energy. Afterwards, the indenter begins to rise. As Figure 5h shows, when the depth decreases to 20.55 Å (corresponding to H in Figure 3b), the prismatic dislocation loop glides toward the lower surface of the substrate along the (111) plane, which may lead to the load relaxation. As the indenter continues to rise, the prismatic dislocation loop begins to move toward the upper free surface because of the free boundary condition; meanwhile, the ($1\bar{1}1$) TB extends downward. In Figure 5i, as the indenter returns to its initial position (corresponding to I in Figure 3b), the prismatic dislocation loop returns to the bottom of the indentation and shrinks, while the ($1\bar{1}1$) TB still exists. Moreover, four vacancies are generated as the results of the dislocation reactions, which can contribute to the reduction of stress; a similar behavior was observed during the nanoindentation of the FeNi_3_ (110) surface, which was considered to be the result of pile-up behavior [30].

#### 3.2.3. Analysis of Defect Evolution for the (111) Sample

As displayed in Figure 6a, as the indentation depth increases to 3.8 Å (corresponding to A in Figure 3c), a dislocation embryo nucleates under the top surface, which will develop into a 1/6[$11\bar{2}$] Shockley dislocation. As the indentation depth increases, dislocations are formed constantly. In Figure 6b, when the indentation depth reaches 6.6 Å (corresponding to B in Figure 3c), SISFs are produced on the ($\bar{1}11$) and ($11\bar{1}$) planes because of dislocation slip, and a 1/3[010] Hirth dislocation is produced by the dislocation reaction 1/6[211] + 1/6[$\bar{2}1\bar{1}$] →1/3[010]. As Figure 6c shows, when the depth is 10.15 Å (corresponding to C in Figure 3c), dislocations continue to increase, an extended dislocation is formed by a 1/2[110] Perfect dislocation dissociating into 1/6[121] and 1/6[$21\bar{1}$] Shockley dislocations that connected by an SISF on the ($1\bar{1}1$) plane. Furthermore, the ($11\bar{1}$) SISF fades away accompanying with the formation of the ($11\bar{1}$) TB (Figure 6c_0_).

In Figure 6d, as the indentation depth increases to 16.95 Å (corresponding to D in Figure 3c), the ($\bar{1}11$) SISF and ($11\bar{1}$) TB expand downward by the dislocation glide (Figure 6d_0_), the ($1\bar{1}1$) SISF disappears accompanying with the generation of the ($1\bar{1}1$) TB and ($1\bar{1}1$) SESF (Figure 6d_1_). What’s more, a prismatic dislocation loop is formed, which was also observed in the nanoindentation of the Al (111) surface [48], while a 1/6[$01\bar{1}$] Stair-rod dislocation is formed by the dislocation reaction 1/6[$\bar{1}21$] + 1/6[$1\bar{1}\bar{2}$] →1/6[$01\bar{1}$] in this study. In Figure 6e, as the depth increases to 20.4 Å (corresponding to E in Figure 3c), due to the various dislocation activities, the ($11\bar{1}$) TB and ($1\bar{1}1$) SESF annihilate, and the ($1\bar{1}1$) SISF expands downward. In addition, the prismatic dislocation loop glides toward the lower surface, and similar behavior was observed during the nanoindentation of the Ni (111) surface [42].

In Figure 6f, as the depth is the maximum 22.6 Å (corresponding to F in Figure 3c), a (111) TB (Figure 6f_0_), two SISF on the ($\bar{1}11$) and ($11\bar{1}$) plane are formed; the prismatic dislocation loop continues to glide and dissociate into two parts adhering to the side surfaces since the periodic boundary condition. However, during the nanoindentation of the Al (111), the prismatic dislocation loop glided downward to the bottom surface and disappeared to leave a step [48]. As Figure 6g shows, when the depth is 18.15 Å (corresponding to G in Figure 3c), the (111) TB and SISFs all annihilate, and the two parts that adhered to the side surfaces move to the top surface with a certain distance because of the free boundary condition. In addition, a stacking fault tetrahedron (SFT) is formed under the indentation, which was observed in the nanoindentation of the FeNi_3_ (111) surface [30]; and five vacancies are formed by the dislocation reactions. In Figure 6h, as the indenter returns to its initial position (corresponding to H in Figure 3c), the SFT and other dislocations annihilate with dislocation slip; while the vacancies increase to seven, and SISFs on the ($\bar{1}11$) and ($1\bar{1}1$) planes are formed.

### 3.3. Analysis of Strain Energy for Different Samples

As is depicted in Figure 7, for each sample, with the increase of step, the strain energy increases to the maximum due to the intensification of plastic deformation; meanwhile, the indentation depth also reaches the maximum. Afterwards, the energy decreases with depth owing to the deformation recovery. It can be seen that the crystal orientation has a significant effect on strain energy. In the plastic deformation stage, the strain energy of the (111) sample is the highest, followed by the ($\bar{1}10$) sample, while the strain energy of the (100) sample is the lowest. These differences are related to the difficulty of activating slip system in different samples. For the (111) sample, the slip system is activated difficultly, the lattice distortion degree is great, hence the strain energy of the system is the highest. For the (100) sample, the slip system is activated easily, the lattice distortion degree is small, hence the strain energy of the system is the lowest.

### 3.4. Calculation of Hardness and Elastic Modulus

In order to further investigate the effect of the crystal orientation on γ-TiAl during nanoindentation, the Oliver–Pharr method [49,50] is employed to calculate the elastic modulus and hardness. The hardness is the resistance of materials to plastic deformation, it can be calculated by Equations (2)–(6):(2)$$H=\frac{P_{\max}}{A_{c}}$$
(3)$$A_{c}=\pi a^{2}=\pi (2Rh_{c}-h_{c}{}^{2})$$
(4)$$h_{c}=h_{\max}-\varepsilon \frac{P_{\max}}{S}$$
(5)$$S={\frac{dp}{dh}|}_{h=h_{\max}}$$
(6)$$P=B{\left(h-h_{f}\right)}^{m}$$
where, $P_{\max}$ is the load at the maximum depth, $A_{c}$ is the contact area, *R* is the indenter radius, $h_{c}$ is the contact depth, $h_{\max}$ is the maximum depth, *ε* is the correction factor, and *ε* = 0.75 for the spherical indenter [51], fitting 25–50% of the incipient unloading curve with Equation (6) and the stiffness *S* is obtained by Equation (5), $h_{f}$ is the indentation depth after unloading, B and m are fitting parameters.

The value of hardness for each sample is listed in Table 2. Obviously, the value of the (111) sample is the maximum, and that of the (100) sample is the minimum. The results indicate that the (100) sample has the weakest resistance to plastic deformation, as the nanoindentation along the [100] orientation is the most beneficial to activating the slip system; however, for the (111) sample, the nanoindentation along the [111] orientation is the most unfavorable for activating the slip system, and the substrate has the strongest resistance to plastic deformation. Note that the hardness of the ($\bar{1}10$) sample is in good agreement with the experimental results 6.8 ± 0.8 GPa [31].

From a macro point of view, the elastic modulus represents the ability of a material to resist elastic deformation. At the micro level, the elastic modulus is a reflection of the bond strength between atoms, ions or molecules. The elastic modulus can be calculated by the reduced elastic modulus $E_{r}$, which takes into account the combined elastic effects of the indenter and substrate as in Equation (7):(7)$$E_{r}=\frac{S\sqrt{\pi }}{2\beta \sqrt{A_{c}}}$$
where *β* is a constant related to the geometrical shape of the indenter, and *β* = 1 for the spherical indenter.

The finally elastic modulus is calculated from Equation (8):(8)$$\frac{1}{E_{r}}=\frac{1-\nu^{2}}{E}+\frac{1-\nu_{i}{}^{2}}{E_{i}}$$
where *E* and $E_{i}$ are the elastic modulus of the substrate and indenter, respectively; ν and $\nu_{i}$ are the Poisson’s ratios of the substrate and indenter, respectively. Due to the indenter is rigid, its elastic modulus is infinite, and its Poisson’s ratio is 0 [26], while the Poisson’s ratio of the substrate is 0.23 [52].

The value of elastic modulus for each sample is listed in Table 2. Clearly, the value of the (100) sample is the minimum, which suggests that the elastic deformation of this case is the most intense; the value of the (111) sample is the maximum, and its elastic deformation is the slightest. The previous experimental results indicated that the elastic modulus of the (110) sample is higher than that of the (100) sample [31], which is similar to our results. The main reason for these differences is that the bond strength of atoms in the (100) sample is the weakest and the cohesive energy is the lowest; on the contrary, the bond strength of atoms in the (111) sample is the strongest, and the cohesive energy is the highest. Moreover, the value of the (100) sample is within the range of 157.2–167.9 GPa obtained in other works [26,53], and the value of the ($\bar{1}10$) sample is consistent with the experimental results 163 ± 8 GPa [31].

## 4. Conclusions

In this paper, the effect of crystal orientation on the deformation mechanisms and mechanical properties of γ-TiAl during nanoindentation at 300 K was demonstrated by molecular dynamics simulations. The load-depth curve, the defect evolution process as well as the related mechanical parameters of different samples were analyzed. In particular, the following conclusions can be drawn:

(1) During the loading process of each case, there is no pronounced pop-in event in the load-depth curve when the initial plastic deformation of γ-TiAl occurs, which can be explained by the dislocation nucleating before the first load drop.

(2) During the unloading process of the ($\bar{1}10$) and (111) cases, there is a peak in both load-depth curves, because several dislocations move to the free surface and annihilate causing the release of energy.

(3) Stacking faults (intrinsic stacking faults and extrinsic stacking faults), twin boundaries and vacancies are generated in all cases. While interstitials are formed only in the (100) sample; a stacking fault tetrahedron is formed in the (111) sample; and a prismatic dislocation loop is formed in both the ($\bar{1}10$) and (111) samples.

(4) For the ($\bar{1}10$) sample, the prismatic dislocation loop moves to the upper free surface and shrinks with the rise of the indenter. However, for the (111) sample, the prismatic dislocation loop continues to glide downward with the indenter returning and finally dissociates into two parts adhering on the side surfaces of the substrate, which is caused by the periodic boundary condition.

(5) The values of the critical load, strain energy, elastic modulus and hardness for the (111) sample are the maximal, and those for the (100) sample are the minimal. These differences are attributed to the bond strength of atoms and the ease of crystal slip in different samples. Besides, the orientation dependence of the elastic modulus is greater than the hardness and critical load.

## Figures and Tables

**Figure 1 materials-12-00770-f001:**
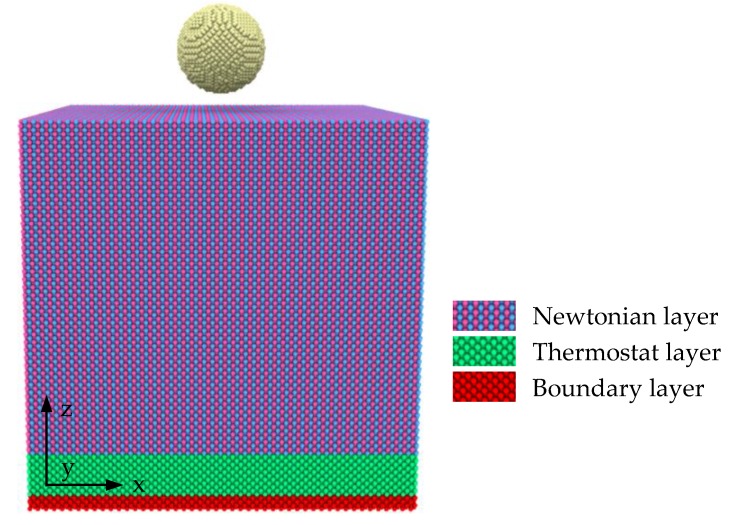
Nanoindentation model of γ-TiAl.

**Figure 2 materials-12-00770-f002:**
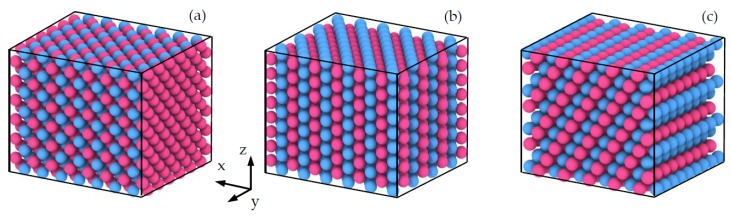
The crystal structure of (**a**) the (100) sample, (**b**) the ($\bar{1}10$) sample and (**c**) the (111) sample. The Ti atoms are colored purple and the Al atoms are colored blue.

**Figure 3 materials-12-00770-f003:**
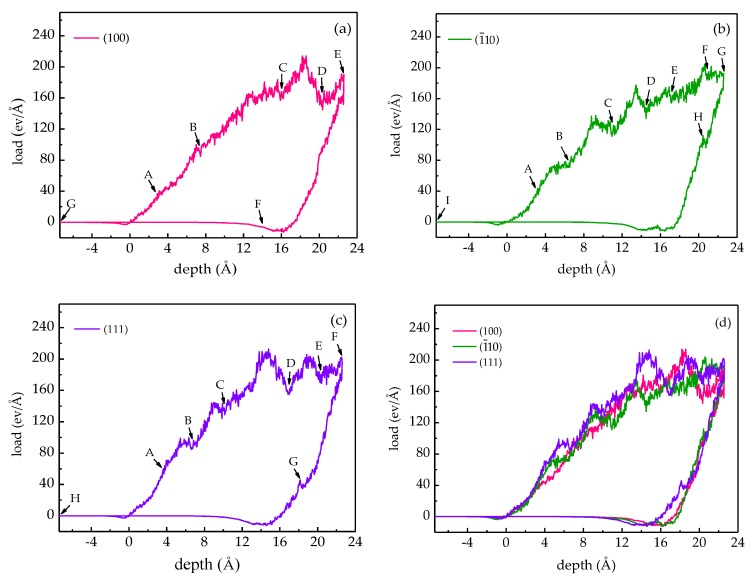
Load-depth curve of (**a**) the (100) sample, (**b**) the ($\bar{1}10$) sample and the (**c**) (111) sample; (**d**) shows the load-depth curves of the three different samples. The letters labeled in (**a**–**c**) represent the characteristic points, and the defect evolution at these points will be described in Section 3.2.

**Figure 4 materials-12-00770-f004:**
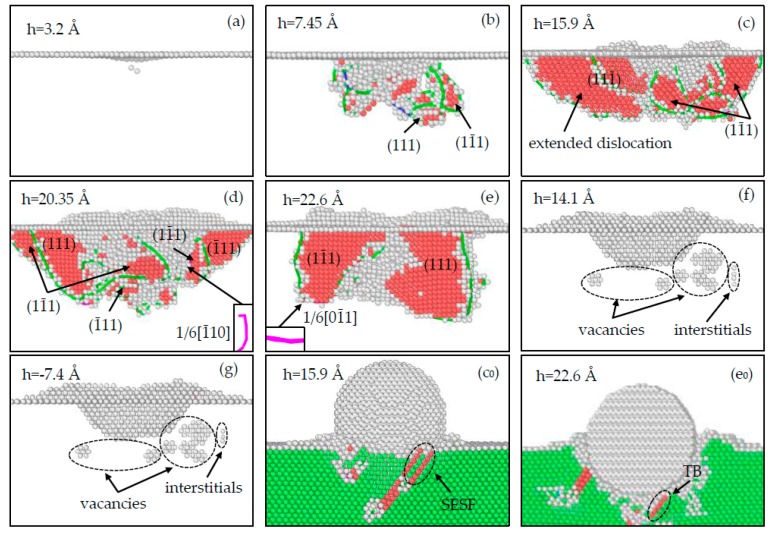
Defect evolution in the (100) sample during the loading process (**a**–**g**,**c_0_**,**e_0_**), and the unloading process (**f**–**g**). (SESF is the external stacking fault, TB is the twinning boundary).

**Figure 5 materials-12-00770-f005:**
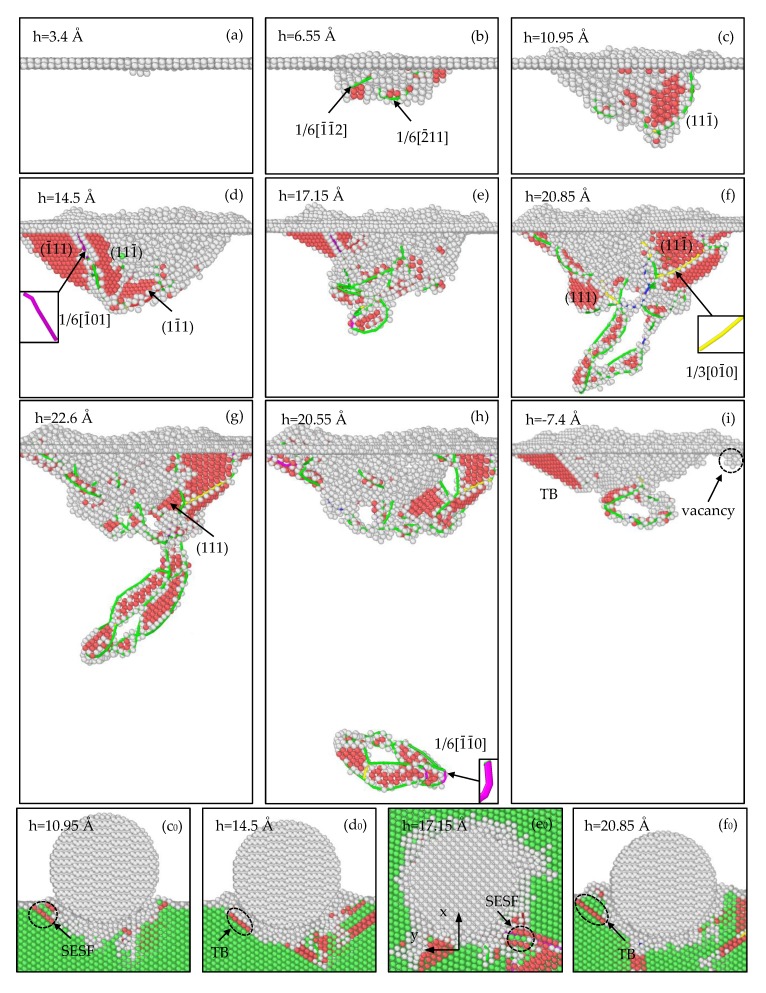
Defect evolution in the ($\bar{1}10$) sample during the loading process (**a**–**g**,**c_0_**–**f_0_**), and the unloading process (**h**–**i**). (SESF is the external stacking fault, TB is the twinning boundary).

**Figure 6 materials-12-00770-f006:**
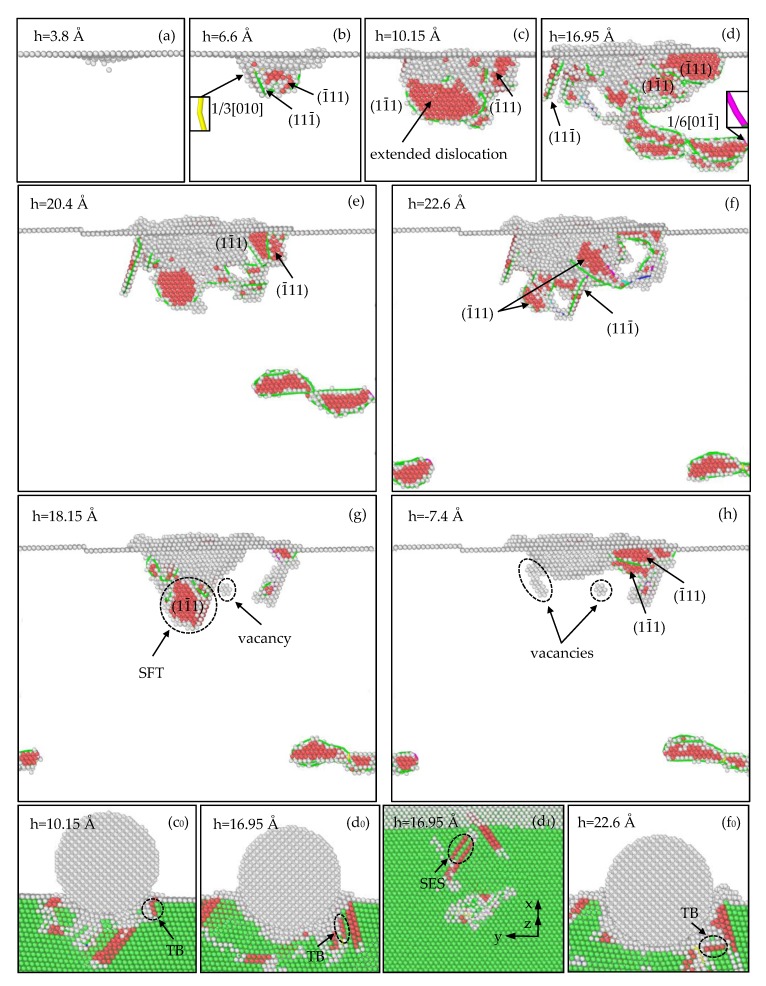
Defect evolution in the (111) sample during the loading process (**a**–**f**,**c_0_**–**f_0_**,**d_1_**), and the unloading process (**g**–**h**). (SFT is the stacking fault tetrahedron.).

**Figure 7 materials-12-00770-f007:**
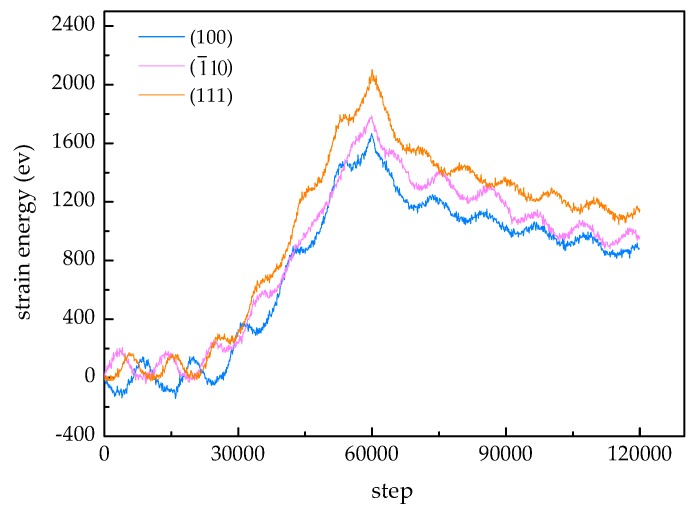
Strain energy-step curves of different samples.

**Table 1 materials-12-00770-t001:** Mie 6–12 (Lennard–Jones) potential function parameters used in simulation.

Parameters	*σ* (Å)	*ϵ* (10^−1^ev)	*r*_0_ (Å)
C-Al	2.976	3.15	7.44
C-Ti	3.759	0.314	9.398

**Table 2 materials-12-00770-t002:** Values of hardness and elastic modulus for different samples.

Samples	(100)	$\left(\bar{1}10\right)$	(111)
Hardness (GPa)	6.42	6.63	6.91
Elastic modulus (GPa)	164.5	175.8	192.4

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
