# Peer review of "Nanoindentation of ?-TiAl with Different Crystal Surfaces by Molecular Dynamics Simulations"

_materials, 2019, doi:10.3390/ma12050770_

Reviewer 1 Report

This manuscript entitled “Nanoindentation of TiAl Alloy with Different Crystal Surface Orientations by Molecular Dynamics Simulation” investigates the effect of crystal orientation on the deformation mechanisms and mechanical properties of TiAl alloy at room temperature during nanoindentation by molecular dynamics. The manuscript is well written and organized. Presented results reveal that  there is no obvious pop-in event in load-depth curves when the initial plastic deformation of TiAl specimens occurs as dislocations nucleates before the first load-drop, while a peak appears in the unloading curves of (1̄10) and (111) specimens due to the release of energy. The orientation dependence of elastic modulus is also investigated.

Nevertheless, I cannot see the importance of the work from the Introduction part. Most of the Refs. are old which means that the subject is not really interesting. Is this a real need to study the system at the chosen conditions? Moreover, molecular dynamics is not really good instrument to study such a complex system in my opinion. At least it should be carefully confirmed that the results are in good agreement with the experiments.

The publication of the present manuscript can be considered after major revision.

Author Response

Point 1: I cannot see the importance of the work from the Introduction part. Most of the Refs. are old which means that the subject is not really interesting. Is this a real need to study the system at the chosen conditions?

Response 1: Thanks for your comment. Considering the problem you pointed out, the introduction has been revised and new references have been added. The revised sentences are marked in red.

Point 2: Moreover, molecular dynamics is not really good instrument to study such a complex system in my opinion. At least it should be carefully confirmed that the results are in good agreement with the experiments.

Response 2: Thank you for your comment. As our introduction state, “It is difficult to investigate the transient atomic information inside the materials during nanoindentation experiments. As a powerful supplement to the experiment, the molecular dynamics method can simulate the atomic scale interaction between the indenter and materials, analyse the evolution laws of defects, and then obtain the mechanical properties and deformation mechanism in detail.”, and there are many studies are based on this method, so it is helpful to investigate the mechanical properties and deformation mechanism of materials.

In the second paragraph of the original section 3.4 “The results are shown in Table 2. The value of (100) specimen is in good agreement with 16.2 GPa in other studies involving experimental results [Dasilva, C.J.; Rino, J.P. Atomistic simulation of the deformation mechanism during nanoindentation of gamma titanium aluminide. Comput. Mater. Sci, 2012, 62:1-5, DOI:10.1016/j.commatsci.2012.04.046.; Han, S.M.; Shah, R.; Banerjee, R.; Viswanathan, G.B.; Clemens, B.M. Combinatorial studies of mechanical properties of TiAl thin films using nanoindentation. Acta Mater, 2005, 53(7):2059-2067.] … …”, the hardness 16.42 GPa obtained from the (100) specimen was compared with the experimental value 16.2 GPa [Han, S.M.; Shah, R.; Banerjee, R.; Viswanathan, G.B.; Clemens, B.M. Combinatorial studies of mechanical properties of TiAl thin films using nanoindentation. Acta Mater, 2005, 53(7):2059-2067.], the results show that they are consistent with each other. In order to highlight the “experiment”, the original sentence has been replaced by “The value of the (100) specimen is in good agreement with the 16.2 GPa in other studies involving experimental results [24,50].”.

Following your advice, we have compared the elastic modulus for the (100) specimen and the experimental results 163±8GPa [M, Göken.; Kempf, M.; Nix, W. D. Hardness and modulus of the lamellar microstructure in PST-TiAl studied by nanoindentations and AFM[J]. Acta. Mater, 2001, 49(5):903-911. DOI10.1016/S1359-6454(00)00375-X.], it was found that the experimental results and ours show good agreement. Meanwhile, we have revised the original sentence to “The value of the (100) specimen is within the range of 157.24~167.94 GPa obtained in other works [25,52], which is consistent with the experimental results 163±8 GPa [53].”

Lastly, we found the previous experiment results showed that the elastic modulus of the (110) specimen is higher than the (100) specimen [Kempf, M.; Goken, M.; Vehoff, H. The mechanical properties of different lamellae and domains in PST-TiAl investigated with nanoindentations and atomic force microscopy. Mater. Sci. Eng. A, 2002, 329-331(none):184-189, DOI:10.1016/S0921-5093(01)01561-1.], which is similar to our results. We have added “… … The previous experimental results indicated that the elastic modulus of the (110) specimen is higher than the (100) specimen [30], which is similar to our results.” in the last paragraph of section 3.4.

In addition to your comments, we have submitted our manuscript to MDPI for English editing, and then the English grammar has been improved and coloured red.

Reviewer 2 Report

This paper details a molecular dynamics study of nanoindentation of TiAl alloys with different surface orientations. The methodology is sound and the material should be of interest to for those specializing in nanoindentation in general or the material system specifically. However, there are some improvements to be made before the manuscript can be published.

As a general comment, the English in the paper needs to be improved. While the text is definitely understandable, it is sometimes difficult to read due to poor grammar or repetitive wording.

The crystal structure of TiAl is not explicitly stated in the text. On a related note, the resolution of Figure 2 is such that is difficult to judge how the atoms are aligned, especially in figure 3c. I suggest supplementing the figure with a panel showing just one unit cell of the crystal.

The authors state that the thermostat layer is 30 Å thick, but they do not state what kind of thermostat is used.

The loading speed used is 50 m/s. Please compare this to some typical values used in experiments.

The authors state that "The initial relaxation was performed properly for 380 ps before indentation". Please clarify what is meant by "properly" in this case.

The load-depth curves in figure 3 are marked with the letters A-G. While the meaning of these letters becomes clear later on in the paper, please add an explanation in the figure caption. I also note that the y-axis is labeled "load" in a,b, and d, but "force" in c. Please be consistent. 

Please explain in a more detail how the critical loads are obtained from the load-depth curves.

In section 3.2, please explain why these particular snapshots were chosen. Based on the load-depth curves, there seems to be other points of interest, for instance the peak loads, which are not investigated.

In the same section, the authors state that the atoms arrange chaotically when deviating from their original positions. Please elaborate what is meant by chaotically if it truly is chaotic in a physical sense. If it is just meant as another way of saying disordered, please rephrase.

As a final note, I would advise the authors to consider that the images showing the various dislocations and stacking faults can be quite difficult to interpret, even with the text as help, as they are quite small and many of the finer details become difficult to see clearly, especially in print. They may want to emphasize these details in some way.

Author Response

Point 1: As a general comment, the English in the paper needs to be improved. While the text is definitely understandable, it is sometimes difficult to read due to poor grammar or repetitive wording.

Response 1: Thanks for your comment. We have submitted the manuscript to MDPI for English editing and improved the English, the revised sentences are highlighted in red in manuscript.

Point 2: The crystal structure of TiAl is not explicitly stated in the text. On a related note, the resolution of Figure 2 is such that is difficult to judge how the atoms are aligned, especially in figure 3c. I suggest supplementing the figure with a panel showing just one unit cell of the crystal.

Response 2: Thanks for your advice, it is our negligence. We have given clear images with the arrangement of atoms, instead of previous Figure 2. As well as the previous Figure 2 caption has been replaced by “The crystal structure of (a) the (100) specimen, (b) the (-110) specimen, (c) the (111) specimen. The Ti atoms are coloured pink and the Al atoms are coloured blue.”.

Point 3: The authors state that the thermostat layer is 30 Å thick, but they do not state what kind of thermostat is used.

Response 3: Thanks for your comment, we are very sorry for our negligence. In our study, the thermostat layer used the velocity scaling method to control the temperature and we have added “using the velocity rescaling method”  in the original sentence.

Point 4: The loading speed used is 50 m/s. Please compare this to some typical values used in experiments.

Response 4: Thank you for your suggestion. In the first paragraph of section 2, we stated that “Owing to the limitation of computing power, the loading speed in molecular dynamics simulation is generally between 1-100 m/s [Jian, S.R.; Fang, T.H.; Chuu, D.S. Nanomechanical characterizations of InGaN thin films. Appl. Surf. Sci, 2006, 252(8):3033-3042. DOI:10.1016/j.apsusc.2005.05.019.]. In our simulations, a constant velocity of 50m/s is adopted along the direction of -z.”, while the loading speed in nanoindentation experiments is about 10-6-10-9 m/s [Liang, H.; Woo, C.H.; Huang, H.; Ngan, A.H.W.; Yu, T.X. Crystalline plasticity on copper (001), (110), and (111) surfaces during nanoindentation[J]. Comp. Model. Eng, 2004, 6(1):105-114. DOI:10.1016/S0960-0779(03)00055-9.; Zhu, P.Z.; Hu, Y.Z.; Wang, H. Atomistic simulations of the effect of a void on nanoindentation response of nickel[J]. Sci. China. Ser. G, 2010, 53(9):1716-1719. DOI10.1007/s11433-010-4094-y.]. It is obvious that the loading speed generally in molecular dynamics simulation is much higher than experiments. However, take into account the simulation time and cost, the speed of 1-100 m/s is widely used in molecular dynamic simulations. Furthermore, the 50 m/s was also applied in other studies [Hu, T.Y.; Zheng, B.L.; Hu, M.Y.; He, P.F.; Yue, Z.F Molecular dynamics simulation of incipient plasticity of nickel substrates of different surface orientations during nanoindentation. Mater. Sci. Technol, 2015, 31(3):1743284714Y.000. DOI: 10.1179/1743284714Y.0000000524.; Fu, T.; Peng, X.; Chen, X.; Weng, S.; Hu, N.; Li, Q.; Wang, Z. Molecular dynamics simulation of nanoindentation on Cu/Ni nanotwinned multilayer films using a spherical indenter[J]. Sci. Rep, 2016, 6:35665. DOI10.1038/srep35665.].

In this paragraph, the original sentence has been replaced by “A loading speed of 50 m/s is applied along the direction of –z, which is much higher than 10-6-10-9 m/s in nanoindentation experiments [36,37]. However, owing to the limitation of computing power, the loading speed in molecular dynamics simulation is generally between 1-100 m/s [38]. Furthermore, the speed of 50 m/s was also chosen in other studies [26, 39].”

Point 5: The authors state that "The initial relaxation was performed properly for 380ps before indentation". Please clarify what is meant by "properly" in this case.

Response 5: Thanks for your advice. The “properly” means that after relaxation of 380ps, it is obvious that the system energy has been stable, as well as the system has been in equilibrium. We have added “to cause the system to be in equilibrium.” in the original sentence.

Point 6: The load-depth curves in figure 3 are marked with the letters A-G. While the meaning of these letters becomes clear later on in the paper, please add an explanation in the figure caption. I also note that the y-axis is labeled "load" in a,b, and d, but "force" in c. Please be consistent. 

Response 6: Thank you for your suggestions, it is our negligence. We have revised the "force" to "load" and added an explanation that “The letters labeled in (a)~(c) represent the characteristic depths, the defect evolutions at these depths will be described in section 3.2.” in the Figure 3 caption.

Point 7: Please explain in a more detail how the critical loads are obtained from the load-depth curves.

Response 7: Thanks for your comment. In this paper, the critical load is the force exerted by the indenter on the specimen when the incipient plastic deformation occurs, due to the initial dislocation nucleation represents the transformation of elastic-plastic deformation, we can obtain the critical depth of the initial dislocation nucleation by the software ovito, and the critical load can be obtained by the critical depth from the load-depth curve. We have added “the transformation of elastic-plastic deformation is the force exerted by the indenter on the specimen when the initial dislocation nucleates, we obtain the critical depth of the initial dislocation nucleation by the Ovito, and the critical load can be obtained by the critical depth from the load-depth curve in Figure 3.” in the last paragraph of section 3.1.

Point 8: In section 3.2, please explain why these particular snapshots were chosen. Based on the load-depth curves, there seems to be other points of interest, for instance the peak loads, which are not investigated.

Response 8: Thanks for your comments. In this section, our main purpose is to describe the defect evolutions of TiAl specimen with different crystal surface orientations. In addition to the content in this section, we have also investigated several points at peak loads, the results showed that the defect evolutions at these valley points is more typical and interesting than those peak points at valley loads, for instance the generation of ESFs and TBs, special dislocation interactions. As well as these points we selected can be enough to demonstrate the difference of defect evolutions in different specimens. Furthermore, the similar points were also chosen in other study [Fu, T.; Peng, X.; Chen, X.; Weng, S.; Hu, N.; Li, Q.; Wang, Z. Molecular dynamics simulation of nanoindentation on Cu/Ni nanotwinned multilayer films using a spherical indenter[J]. Sci. Rep, 2016, 6:35665. DOI10.1038/srep35665.]. Therefore, we chosen these snapshots in section 3.2.

Point 9: In the same section, the authors state that the atoms arrange chaotically when deviating from their original positions. Please elaborate what is meant by chaotically if it truly is chaotic in a physical sense. If it is just meant as another way of saying disordered, please rephrase.

Response 9: Thanks for your advice, we are very sorry for the confusing statement. In this paper, the word “chaotically is just meant as another way of saying disordered, we have replaced it by “disordered”.

Point 10: As a final note, I would advise the authors to consider that the images showing the various dislocations and stacking faults can be quite difficult to interpret, even with the text as help, as they are quite small and many of the finer details become difficult to see clearly, especially in print. They may want to emphasize these details in some way.

Response 10: Thanks for your comment, considering the layout and integrity of the images, a certain magnification of the images has been made, and important dislocations have been extracted. What’s more, the Figure 1 has been replaced by a higher-resolution figure.

Round  2

Reviewer 1 Report

Manuscript was improved by the Authors and now can be accepted in the present form.

Author Response

Thank you very much for your attention and suggestions to our manuscript, and we have made a minor revision to the introduction.